# The In Vivo and In Vitro Assessment of Pyocins in Treating *Pseudomonas aeruginosa* Infections

**DOI:** 10.3390/antibiotics11101366

**Published:** 2022-10-07

**Authors:** Abdulaziz Alqahtani, Jonathan Kopel, Abdul Hamood

**Affiliations:** 1Clinical Laboratory Sciences, College of Applied Medical Sciences, King Khalid University, Abha 61321, Saudi Arabia; 2Department of Immunology and Microbiology, Texas Tech University Health Sciences Center, Lubbock, TX 79763, USA

**Keywords:** *Pseudomonas*, pyocin, virulence factors, treatment, infection

## Abstract

*Pseudomonas aeruginosa* can cause several life-threatening infections among immunocompromised patients (e.g., cystic fibrosis) due to its ability to adapt and develop resistance to several antibiotics. In recent years, *P. aeruginosa* infections has become difficult to treat using conventional antibiotics due to the increase multidrug-resistant *P. aeruginosa* strains. Therefore, there is a growing interest to develop novel treatments against antibiotic-resistance *P. aeruginosa* strains. One novel method includes the application of antimicrobial peptides secreted by *P. aeruginosa* strains, known as pyocins. In this review, we will discuss the structure, function, and use of pyocins in the pathogenesis and treatment of *P. aeruginosa* infection.

## 1. General Characteristics of *Pseudomonas aeruginosa*

*Pseudomonas aeruginosa* is a Gram-negative rod-shaped bacterium that can grow in both aerobic and anaerobic conditions, which allows the bacteria to grow in the soil, water, plants, and humans [1,2,3,4]. *Pseudomonas aeruginosa* requires only minimal nutritional supply to grow in several physical conditions, such as high concentration of salts, dyes, and media containing a wide range of antibiotics [4]. *P. aeruginosa* has a large, complex genome (5–7 Mb) encoding several genes and regulatory networks involved in utilizing several carbon sources depending on the environment [4,5]. As a result, *P. aeruginosa* causes several life-threatening acute and chronic infections among immunocompromised patients, such as cystic fibrosis [4]. The ubiquitous clinical presence of *P. aeruginosa* is attributed to its ability to adapt and develop resistance to several antibiotics [4]. In recent years, *P. aeruginosa* infections have become difficult to treat using conventional antibiotics (aminoglycosides, cephalosporins, fluoroquinolones, and carbapenems) due to the increase multidrug-resistant *P. aeruginosa* strains worldwide [4]. For example, 13% of all *P. aeruginosa* infections in the United States are caused by multidrug resistant *P. aeruginosa* strains leading to increased morbidity and mortality [4]. Therefore, there is a growing interest to develop novel treatments for antibiotic-resistance *P. aeruginosa* strains. In this review, we discuss the structure, function, and use of pyocins in the pathogenesis and treatment of *P. aeruginosa* infections.

## 2. *P. aeruginosa* Virulence Factors

### 2.1. Cell-Associated Virulence Factors

The pathogenesis of *P. aeruginosa* is mediated by its wide range of virulence factors, which include cell-associated and extracellular virulence factors [6]. Cell-associated virulence factors include polar flagellum, type IV pili, lipopolysaccharides (LPS), and alginate [7,8]. The single polar flagellum of *P. aeruginosa* mediates swimming and swarming motility, while the type IV pili facilitates twitching motility [6]. The flagella are essential for *P. aeruginosa* quorum sensing, which is the ability to detect and respond to cell population density by gene regulation [4]. This quorum sensing allows *P. aeruginosa* to colonize a host in acute or chronic infections through modifications to quorum sensing genes [4]. Specifically, 10% of all *P. aeruginosa* genes involved in virulence factor production, motility, motility-sessility switch and biofilm development are regulated by alterations in quorum sensing [4]. Both flagella and pili assist in the adherence of *P. aeruginosa* to host tissues, and elicit inflammatory responses through interacting with Toll-like receptor 5 (TLR5) and 2 (TLR2), which increases IL-8 expression [7,9,10]. In contrast, lipopolysaccharide (LPS) is a major molecule of the outer membrane of Gram-negative bacteria, consisting of three components: a membrane-anchored lipid A, a polysaccharide core region, and a highly variable O-antigen [11]. When present in the blood, LPS triggers pulmonary inflammation through increasing the levels of proinflammatory cytokines [12].

In addition, the Lipid A forms the hydrophobic portion of LPS and helps anchor molecules to the outer member. Interestingly, the number of acyl chains, degree of phosphorylation, and presence of other post-translational modifications have important functions in the pathogenesis of *P. aeruginosa* [13]. Specifically, the different lipid A isoforms have different potencies activating the host immune system through binding to TLR4 [13,14]. The O-antigen of *P. aeruginosa* LPS consists of long polysaccharide that is responsible for conferring serogroup specificity [14]. Lastly, alginate is a mucoid exopolysaccharide (EPS) produced by specific *P. aeruginosa* strains typically isolated from the lungs of cystic fibrosis patients [7]. Alginate also protects *P. aeruginosa* from host immune systems through enhancing adhesion a to solid surfaces [15]. Furthermore, the alginate is an essential component for developing *P. aeruginosa* biofilms and the main constituent of the glycocalyx [15]. The alginate also protects *P. aeruginosa* from the host immune system through reducing free radicals released by activated macrophages, thereby reducing phagocytosis [16]. Therefore, the flagella, LPS, and alginate are essential components for establishing *P. aeruginosa* during the initial stages of infection. Furthermore, extracellular virulence factors, known as bacteriocins/pyocins, are an essential for the pathogenesis of *P. aeruginosa*.

### 2.2. Extracellular Virulence Factors

*Pseudomonas aeruginosa* produces various extracellular virulence factors including exotoxin A (ETA), exoenzymes, proteases, pyocyanin, and siderophores [17]. Exotoxin A (ETA), the most virulence factor produced by *P. aeruginosa*, is an ADP-ribosylating enzyme that modifies host elongation factor-2 thereby inhibiting the protein synthesis process, leading eventually to cell death [7,18]. In addition, ETA induces apoptosis, or the programmed death of host cells. ETA is secreted by type II secretion system (T2SS) [19]. Exoenzymes are effector proteins of the type III secretion system (T3SS). T3SS is a needle-like apparatus that injects toxins directly into host cells [20]. Exoenzymes include ExoS, ExoU, ExoT, and ExoY [21]. ExoS and ExoT are ADP-ribosylating enzymes that disrupt actin filaments of the cytoskeleton. ExoT induces mitochondrial apoptosis in host cells. ExoU is a major cytotoxin that exhibits a phospholipase activity and facilitates phagocyte’s killing [21]. The cytotoxic effect of ExoU is 100 times higher than that of ExoS [22]. ExoY is an adenylate cyclase that disrupts actin filaments of the cytoskeleton, and increases the production of the second messengers cGMP and cUMP in host cells [23]. 

*P. aeruginosa* secretes several proteases including LasA, LasB, alkaline protease, and protease IV [7,8]. LasA is staphylolytic zinc metallopeptidase with elastolytic activity. LasB is zinc metalloprotease exhibiting the most elastolytic activity, comparing to LasA. The two elastases play an important role in *P. aeruginosa* pathogenesis via degrading blood vessel and the elastin within lung alveoli tissues. Similar to ETA, LasA, and LasB are secreted by the type II secretion system (T2SS). This system is also regulated by the complex quorum-sensing (QS) system, which includes the las, rhl, and pqs [24,25]. Quorum-sending is a cell density dependent communication system between microorganisms to coordinate the production of different virulence factors. Each of the well characterized las and rhl systems consist of three major components: an autoinducer synthase (LasI or RhlI), an autoinducer (N-(3-oxodo-decanoyl)-homoserine lactone (C12-HSL) or N-butyryl-L-homoserine lactone (C4-HSL) and a transcriptional regulator (LasR or RhlR). The autoinducer C12-HSL is synthesized by LasI, while C4-HSL is synthesized by RhlI. When the autoinducer, C12-HSL or C4-HSL reaches a threshold, it binds to the transcriptional regulator LasR or RhlR, respectively. Upon binding with its transcriptional regulator, the complex activates the expression of genes coding virulence factors, including type II secretion system proteins, alkaline protease, alginate, pyocyanin and pyoverdine, elastase, and ExoA [25]. The third QS system, pqs, is mediated by an autoinducer 2-heptyl-3-hydroxy-4-quinolone (Pseudomonas quinolone signal or PQS). Unlike a single autoinducer synthase of the two QS systems, las and rhl, PQS is synthesized by multiple proteins encoded by pqsABCD and pqsH [26]. Alkaline protease is zinc metalloprotease that degrades host immune complements and cytokines such as C1q, C2, C3, IFN-γ, and TNF-α [27]. Protease IV is also capable of degrading host immune complements, clotting factors and surfactant proteins [28].

*Pseudomonas aeruginosa* also produces different molecules include pyocyanin, and siderophores such as pyoverdine and pyochelin. Pyocyanin (PCN), a blue–green pigment, is responsible for the distinct color of *P. aeruginosa* colonies. Pyocyanin causes multiple effects on host cells include initiating oxidative stress via generating reactive oxygen species (ROS), disrupting mitochondrial electron transport, and inducing neutrophils apoptosis [29]. Specifically, in vivo analysis showed that PCN contributes to the pathogenesis of *P. aeruginosa* [29]. Using a mouse model, purified PCN induced bronchopneumonia with neutrophil influx, causing small-airway congestion in the mouse lungs [29]. Siderophores are low molecular weight iron-chelating compounds that scavenge iron from the environment. *P. aeruginosa* secrets two major siderophores; pyoverdine and pyochelin. Pyoverdine is a green fluorescent compound with a higher affinity for iron compared to pyochelin [30]. In vivo analysis, using the murine model of thermal injury and the murine model of lung infection, showed that pyoverdine has a critical role for the survival of *P. aeruginosa* during infections [31]. 

### 2.3. Pseuodomonas aeuroginosa Pyocins

Pyocins are bacteriocins that are narrow spectrum antimicrobial peptides produced by many Gram-negative and positive bacteria. Bacteriocins are antibacterial peptides synthesized that are toxic to closely related bacteria [32]. *Pseudomonas aeruginosa* produce bacteriocins, known as pyocins, which are narrow-spectrum antimicrobial agents that are lethal to closely related bacteria [32]. Specifically, *P. aeruginosa* produce three major pyocins, R, F, and S types pyocins, that vary in their mechanism and structure (Table 1).

### 2.4. The R-Pyocins

In 1964, Kageyama et al. described the first R-pyocins as high-molecular-weight proteins that resemble the nonflexible, contractile tail structures of bacteriophages of the Myoviridae family [33]. Specifically, the R-pyocins is a double hollow cylinder (120 nm long and 15 nm wide) consisting of a rigid core, sheath, baseplate, and tail fibers [3,32,33]. The core is surrounded by a contractible outer sheath and attached to the sheath and baseplate [3,32,33]. The tail fibers are responsible for the attachment of the pyocins to the target cell [32]. Analysis of the tail fiber (Prf15) amino acid sequence, R-pyocins are classified into five major groups: R1-R5. The amino acid sequence of Prf15 of R2, R3, and R4 is nearly identical, while R1 and R5 show differences in the C-terminal region of the pyocin structure [34]. Once the mode of action of different R pyocins are attached to the target membrane, the R-pyocins sheath contracts to insert the pyocin core into the cell envelope, forming a channel that depolarizers the cell membrane, which arrests protein and nucleic acid synthesis [32]. 

### 2.5. The F-Pyocins

The second type of phage tail-like bacteriocins are the F-pyocins. In 1967, Takeya et al. described the structure of F-pyocins, which consisted of flexuous rod-like particles that are composed of a core, a baseplate, and tail fibers containing several short and long filaments [35]. The F-pyocin rod particles are shorter than R-pyocins with an estimated length of 106 nm and width of 10 nm; the length of the tail fibers is also estimated at 50 nm [32,36]. Specifically, one end of the F-pyocin rods form a square at one end that tapers off to form fine fibers estimated at 43 nm in length [3]. To date, three F-subtypes (F1, F2, and F3) have been described [32,36]. The shapes and sizes of these F-pyocin subtypes are similar to R-pyocins in their structure and activity [32]. The filaments are responsible for the binding of F-pyocin onto target cells, which can be modified through different short and long filament compositions [3]. Genes that constitute the R2-F2 loci and their location on the chromosome of *P. aeruginosa* strain PA01 are illustrated in Figure 1. 

### 2.6. The S-Pyocins

*Pseudomonas aeruginosa* produces a various number of colicin-like bacteriocins, known as S-pyocins [3]. Compared to R- and F-pyocins, S-pyocins are water-soluble, heat, and proteases sensitive proteins [3]. To date, several S-pyocin subtypes have been discovered. S-pyocins are simple bacteriocins consisting of large and small protein complexes [3]. The large protein harbors the killing activity through DNA degradation whereas the small protein, known as the immunity protein, is responsible for the protection against the host antibacterial activity. This protection is attributed to the interaction between the C-terminal of the large protein and the N-terminal of the immunity protein [3,32]. In most S pyocins, four domains constitute the large protein arranged from N-terminus to C-terminus: the receptor-binding domain (I), translocation domain (II), domain with an unknown function (III), and killing domain (IV). In contrast, the S1 and S5 pyocins have only three domains (a receptor-binding, translocation, and killing domains) [3,32]. The C-terminal killing domains of S1 and S2 are highly conserve whereas the S1 and S2 are almost identical [37]. The N-terminal receptor-binding domain is less conserved among S-pyocins, which reflects the diversity of receptor recognition. The molecular weight of S-pyocins, particularly the large domains, differ widely ranging between 32–84 kDa; in contrast, the small proteins have a similar molecular weight of 10 kDa [37].

### 2.7. R-, F-, and S-Pyocins Differ in Their Structure and Mode of Action

Although pyocin function remains an active area of investigation, current scientific investigations suggest pyocins have two primary functions in *P. aeruginosa*: protect against different *P. aeruginosa* strains and attack other bacterial species [3]. The R-pyocins are able to eliminate Gram-negative bacteria by recognizing different receptor sites within LPS molecule [3,38,39]. Specifically, Kohler et al. showed that L-rhamnose residues distributed around the outer core of LPS are the receptor target for R1-pyocin, while D-glucose residues are the receptor sites for R2- and R5-pyocins (Figure 1) [39]. Furthermore, R-pyocin tail fibers mediate the recognition and binding to susceptible bacterial surface [3]. Upon binding, the pyocin sheath contracts thereby allowing the core to penetrate through both outer and cytoplasmic membranes [3]. Subsequently, the pyocin core depolymerizes the membranes through leakage of intracellular ions. This process initiates a cascade of cellular activity leading to bacterial death [3,40]. The antibacterial activity of R-pyocin is very efficient process with only a single R-pyocin particle required to eliminate a bacterial cell [41]. Similar to R-pyocins, F-pyocins eliminate their bacterial cells via a single-hit process [3,32,36,42]. However, the killing mechanism of F-pyocins remains unclear. Unlike R-pyocins, F-pyocins possess a flexible core with no contractile sheath, which may not penetrate bacterial cell envelope. Further scientific investigation is required to understand the mechanism behind F-pyocins antibacterial activity [41]. 

In contrast, S-type pyocins destroy bacteria through several different mechanisms (Table 2). For S1-, S2-, S3-, and AP41-pyocins, the killing domains exhibit DNase activity while S4- and S6-pyocins destroy bacteria by inhibiting protein biosynthetic machinery via tRNase or rRNase; in contrast, S5 pyocins destroys susceptible bacteria through pore formation, resulting in membrane damage and leakage of intracellular materials including nucleic acids, and ATP [32,43,44]. The pyocin DNase activity has a conserved HNH-endonuclease motif, which serves as the core of the catalytic site of the endonuclease [32]. The HNH endonuclease motif can also chelate a single metal ion required for hydrolysis of double stranded DNA [32]. The S1-, S2-, and S3-pyocins contains a cognate immunity protein that degrades chromosomal DNA and inhibits lipid biosynthesis in target bacterium. The combination of DNase and lipid II inhibition activities explains the high toxicity of S1-, S2-, and S3-pyocins to bacteria [37]. Pyocin AP41 (also called Ar41) is largest S-pyocin with DNase activity. Unlike S1 and S2, AP41 does not inhibit lipid synthesis of susceptible cells. As a result, the S1 and S2-pyocins have a higher killing efficiency than AP41 [37].

### 2.8. The Antibiofilm Effect of P. aeruginosa Pyocins

*Pseudomonas aeruginosa* remains a difficult bacteria to treat with standard antibiotic treatment due to its ability to form multicellular biofilms, which are a collection of surface-associated microbial cells enclosed in an extracellular polymeric matrix [45]. These biofilms are composed noncellular materials such as mineral crystals, corrosion particles, clay or silt particles, or blood components [45]. In most instances, biofilms are composed of 50–90% organic carbon compounds, such as uronic acids (such as D-glucuronic, D-galacturonic, and mannuronic acids) or ketal-linked pyruvates [45]. These substances bind to divalent cations, such as calcium and magnesium, which provide greater binding force to develop biofilms [45]. As a result, biofilms contribute to the antimicrobial resistance of *P. aeruginosa* by reducing the transport of antibiotics through the biofilm [45]. Interestingly, R-pyocins have been shown to influence strain dominance between communities of *P. aeruginosa* in the presence of biofilms [46]. Although pyocins are effective antimicrobial agents, few studies assessed the efficacy of pyocins in treating *P. aeruginosa* biofilms in vitro. 

A recent study by Smith et al. compared the anti-biofilm efficacy of S2-pyocin and antibiotics against *P. aeruginosa* biofilms in vitro [47]. The study aimed to determine whether the S2-pyocin exhibited antibacterial activity against clinical isolate of *P. aeruginosa* growing in the biofilm state [47]. Specifically, the study treated biofilm of *P. aeruginosa* strain YHP14 for 1 h using pyocin S2, aztreonam, or tobramycin [47]. It was found that the S2-pyocin showed the greatest efficacy against *P. aeruginosa* biofilms showing approximately a 4 log reduction in *P. aeruginosa* survival [47]. A recent study by Paškevičius et al. assessed the activity of plant pyocins, such as the S5-pyocin, against *P. aeruginosa* biofilms in vitro [48]. Similar to the previous study, a biofilm consisting of *P. aeruginosa* strain A19 were treated with 10 µg/mL of different plant pyocins (S5, PaeM, L1, L2, L3, and PaeM4) [48]. Among the pyocins tested, the S5-pyocin displayed the greatest activity against strain A19 biofilm decreasing the viable bacterial cells for 3 logs [48]. In contrast, the PaeM4 pyocin showed a limited effect on strain A19 biofilm with only 2 logs reduction, while the L2- and L3-pyocins showed no effect on the viability of *P. aeruginosa* [48]. Oluyombo et al. also examined the anti-biofilm efficacy of R1- and R2-pyocins against *P. aeruginosa* biofilms in vitro [46]. The biofilms were developed using the *P. aeruginosa* A026 or A018 strains on polypropylene beads, which were subsequently treated with R2 and R1-pyocins [46]. Compared to the control, the R-type pyocins showed a potent activity on the biofilm development by these two strains, reducing cell viability for approximately 3–6 logs [46]. Although the pyocins reduced the cell viability of *P. aeruginosa* biofilms, further clinical studies are needed to evaluate the efficacy and safety of pyocins in patients. 

### 2.9. Assessing the Therapuetic Potential of P. aeruoginosa Pycocins Using Different In Vivo Models

As shown in Table 3, using different in vivo models, several studies examined the efficacy of *P. aeruginosa* pyocins as a prophylactic or therapeutic agent. In 1969, Bird et al. was the first to test pyocins as a therapeutic agent against *P. aeruginosa* infection [49]. In the study, crude pyocins were shown to be protective effects in chick embryos infected with *P. aeruginosa* increasing the survival rate from 3% (untreated-chick embryos) to 55% (pyocin-treated chick embryos) [49]. However, the crude pyocins were shown to be slightly toxic, accounting for 11% mortality in pyocin-treated chick embryos compared to only 6% mortality in nutrient broth-treated chick embryos [49]. Since the tested pyocins were not fully purified, it is likely that the increased toxicity could be due to impurities or interactions between different pyocins [49]. A subsequent study by Merrikin and Terry tested the efficacy of purified pyocin (known as Pyocin 78-C2) in protecting mice from a systemic *P. aeruginosa* infection [50]. In this study, mice were infected intravenously with *P. aeruginosa* and then treated with either an intraperitoneal or intravenous pyocin 78-C9 up to 24 h post-infection [50]. Pyocin 78-C2 rescued infected mice when only administrated intravenously at either 0 or 6 h post-infection [50]. Pyocin 78-C2 had no toxic effects when administrated either intravenously or intraperitoneally [50]. However, it was unknown whether prophylactic administration of pyocin would improve outcomes in the presence of systemic *P. aeruginosa* infection.

Haas et al. examined the prophylactic effect of pyocin in mice using a crude pyocin extracted from *P. aeruginosa* P10 strain was utilized [51]. Mice were injected with a single or multiple intraperitoneal doses of the crude pyocin in the pyocin-sensitive *P. aeruginosa* P10 strain [51]. The study showed pyocins protected against *P. aeruginosa* infection when administrated up to 4 days before symptoms presented; the crude pyocins pyocin had no toxic effects in mice [51]. However, the crude pyocins failed to protect mice when administrated after *P. aeruginosa* infection. The first study to utilize R-, F-, and S-pyocin subtypes for therapeutic applications against *P. aeruginosa* was conducted by Rosamu et al. [52]. The study examined which pyocin subtype was effective at treating *P. aeruginosa* infections [52]. Mice were injected intraperitoneally with R-, F-, or S-type pyocins at different time points [52]. The study showed the R- and F-pyocin subtypes were shown to be protective when administrated before or in conjunction with a lethal infective dose of *P. aeruginosa* [52]. However, the S-pyocin exhibited no protection against *P. aeruginosa* infection in mice [52].

Similar to previous studies using crude pyocin extracts, R- and F-type pyocins showed no adverse effect in mice [52]. The R- and F-pyocins were also tested in a mouse burn model in which burned mice were infected with *P. aeruginosa* strain P14 and then treated topically with either the R- or F-pyocin subtypes [52]. Neither pyocin R- nor F-type was effective in treating *P. aeruginosa* infection in burned mice; specifically, the mortality rate was 60% in pyocin-treated mice, while 70% in untreated mice [52]. Therefore, further research was needed to determine which specific R- or F-pyocin subtypes would improve mortality in the presence of *P. aeruginosa* infection.

A subsequent study by Scholl and Martin examined the antibacterial efficacy of R2-pyocin in *P. aeruginosa* infection in mice [34]. The mice were intraperitoneally injected with a lethal dose (LD90 of ~106 colony formation units (CUFs)) of an R2-sensitive *P. aeruginosa* 13s strain [34]. A single dose of the purified R2-pyocin rescued mice against a lethal *P. aeruginosa* infection, increasing the survival rate to 90–100% after 48 h [34]. The R2-pyocin also showed protective effects when administrated up to 4 h post-infection [34]. A single intravenous dose of R2-pyocin was found to produce neutralizing antibodies in mice, causing a notable reduction in the efficacy of R2-pyocin treatment in mice previously infected and treated by pyocin [34]. Specifically, 60–90% of re-infected and retreated mice died with four of the five decreased mice in the second pyocin treatment developing neutralizing antibodies against the R2-pyocin [34]. Therefore, it is possible that administering pyocins repeatedly in humans infected with *P. aeruginosa* may showed reduced efficacy after repeated use. A similar study McCaughey et al. also showed that pyocins produce different immunogenic responses [53]. Repeated exposure of the S5-pyocin administrated either intranasally or intraperitoneally in mice produced a small amount of S5-specific IgG [53]. However, S5-specific IgG did not interfere with the bactericidal activity of S5-pyocin S5 [53].

Another study by Six et al. examined whether pyocin treatment would be an effective treatment in a murine model in a murine model of sepsis with *P. aeruginosa* [54]. Specifically, the authors used recombinant pyocins S5 and AP41 in both the Galleria mellonella infection model and a murine model of *P. aeruginosa* sepsis. The study showed that when administered to mice alone, neither pyocin had any negative side effects and, instead, showed a strong in vitro anti-pseudomonal activity. Both pyocins S5 and AP41 greatly increased survival in an invertebrate sepsis model using G. mellonella, increasing survival from 10% in controls to 80–100% survival among groups of pyocin-treated larvae [54]. Both pyocins showed extensive organ dispersion following injection into mice. In the murine sepsis model, Pyocin S5 was administered 5 h after infection, which significantly increased survival from 33% to 83%. Overall, the two types of *P. aeruginosa* infection models showed pyocins S5 and AP41 exhibit in vivo biological activity that may increase survivability for multi-drug resistant *P. aeruginosa* infections [54].

Pyocins activity against *P. aeruginosa* activity was also investigated in a nonvertebrate host, Galleria mellonella caterpillar. Smith et al. assessed the ability of recombinant S2 pyocin in protecting *G. mellonella* larvae against a lethal *P. aeruginosa* infection by inoculating larvae with a lethal dose (104 CFU) of *P. aeruginosa* strain YHP14 [47]. After which, the larvae were injected with a 27 mg/kg of the S2-pyocin. The study found that the S2-pyocin protected the *G. mellonella* larvae against a lethal *P. aeruginosa* infection with a survival rate of 100% among pyocin-treated larvae compared to 0% in untreated larvae [47]. In addition, the S2-pyocin was not toxic in the larvae. A similar study by Paškevičius et al. examined the efficacy of pyocins PaeM4, S5, L2, and L3 in the G. mellonella caterpillar [48]. Larvae infected with a lethal dose (500 CFU) of *P. aeruginosa* strain A19 were injected with a 10 µg of either PaeM4-, S5-, L2-, or L3-pyocins [48]. Although there were variations in protective effects of each pyocin against a lethal *P. aeruginosa* infection, the S5-pyocin showed the highest efficacy compared to other tested pyocins, protecting all larvae from the infection [48]. In contrast, Pyocins PaeM4 and L2 rescued 90% and 75% of larvae from the infection, respectively. L3 had no protective effect against *P. aeruginosa* infection in the larvae [48]. In the same study, a combination of pyocins showed a better protective effect against a lethal *P. aeruginosa* infection in larvae than a single pyocin-treatment [48].

## 3. Engineered Pyocins with Broad Antimicrobial Activity

Pyocins have a narrow bactericidal activity targeting only strains within the species of the producer bacteria. This is a useful feature with regard to keeping an intact beneficial flora. However, this feature can prevent pyocins from having a broader antimicrobial activity to other pathogens besides *P. aeruginosa*. A recent study by Ritchie et al. overcame this limitation by engineering the natural spectrum of pyocins [55]. The R2-pyocin was engineered to target *E. coli* O157: H7, a major cause of food-borne colitis and diarrhea through fusing the tail spike from bacteriophage phiV10 to the R2-pyocin tail fiber (known as AvR2-V10.3) [55]. The therapeutic use of AvR2-V10.3 was examined in the infant rabbit model through administering AvR2-V10.3 orogastrically either before or after the onset of diarrhea in rabbits infected with *E. coli* O157: H7 [55]. Specifically, the administration of AvR2-V10.3 before the onset of symptoms prevented diarrhea and reduced the number *E. coli* cells in the intestines and stool (3–5 logs reduction) [55]. AvR2-V10.3 was also effective when administrated after the onset of diarrhea (3 days post-infection), reducing the severity of *E. coli* infection and bacterial load (1–2 logs reduction) in the intestines [55]. Additionally, histological analysis showed that AvR2-V10.3 reduced intestinal inflammation [55]. Therefore, it is possible the pyocin antimicrobial activity may be used for other bacterial infections.

## 4. Advantage of Pyocins in Treating *P. aeruginosa* Infections

Around 1.4 million individuals are hospitalized every year in the U.S. [56,57]. Given the increase in antibiotic-resistant bacteria in hospitals, patients are at high risk to develop nosocomial or health care-associated infections [58,59]. Systemic infections caused by invading bacterial pathogens have a high mortality rate, frequently due to a dysregulated host immune response known as sepsis [60,61]. *Pseudomonas aeruginosa* is one of the leading causes for nosocomial infections with concomitant bloodstream infection and sepsis in hospitalized patients [62,63,64,65,66,67,68]. In recent years, accumulating insights into *P. aeruginosa* virulence factors have revealed new therapeutic targets for *P. aeruginosa* infections. These targets included structural components, such as the pili and flagellum, which are important in bacterial adhesion and translocation, and lipopolysaccharides (LPS), a major surface-associated virulence factor. Additionally, the type III secretion system, which *P. aeruginosa* uses to translocate damaging toxins into host cells, and the quorum sensing system were identified as targets. 

Bacteria frequently manufacture protein antibiotics called bacteriocins for intraspecies competition [69]. Due to their potency and precise targeting, bacteriocins such as pyocins have the potential to be turned into therapeutically relevant antibiotics for Gram-negative bacteria that are notoriously difficult to treat. In *P. aeruginosa*, bacteriocins known as pyocins exhibit strong efficacy in a murine model of *P. aeruginosa* lung infection. For example, the concentration of pyocin S5 needed to provide protection from a lethal infection is at least 100 times lower than that of the most widely used inhaled antibiotic, tobramycin. Additionally, even at high concentrations, pyocins are not immunogenic and their effectiveness is sustained in the presence of pyocin-specific antibodies [69]. Since bacteriocin-encoding genes are widely discovered in microbial genomes, this means that a ready supply of antibiotics with high specificity and potency that are active against antibiotic resistant Gram-negative infections, such as *P. aeruginosa*. Many of these identified targets have been successful enough in enhancing survival and decreasing *P. aeruginosa*-related pathologies in the animal models to be tested further in clinical trials. Although no clinical studies have examined the effect of pyocins in hospitalized patients, the pre-clinical data suggest pyocins may be an effective alternative for the treatment and prevention of *P. aeruginosa* infections. Further randomized clinical trials and prospective studies are needed to evaluate the efficacy and safety of pyocins in hospitalized patients. However, the main advantage of pyocins over other conventional therapies (e.g., antibiotics and bacteriophages) as potential therapeutic agents is their difference in their mode action and their receptors. Utilizing two or more pyocins in treating *P. aeruginosa* infections will significantly reduce the possible emergence of anti-pyocin resistant *P. aeruginosa* mutants.

## Figures and Tables

**Figure 1 antibiotics-11-01366-f001:**
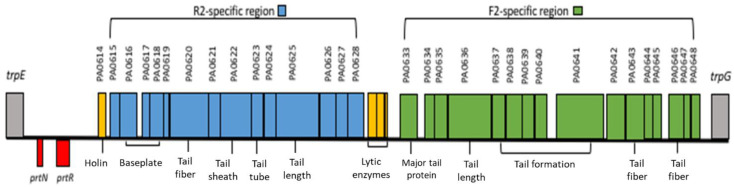
The R2/F2 pyocin gene locus of PAO1 genome. The R2/F2 genes are located between trpE and trpG. Red boxes and yellow boxes represent regulatory genes and lysis gene cassette, respectively. Blue boxes and green boxes represent R2 and F2 genes, respectively (https://www.pseudomonas.com/feature/show?id=103967, accessed on 10 January 2022).

**Table 1 antibiotics-11-01366-t001:** Mechanism of action and structures of different *P. aeruginosa* Pyocins.

Pyocin	Mechanism of Action	Structure	Protease Sensitive
R1-5	Depolarizers cell membrane leading to arrest of protein and nucleic acid synthesis	Rod-like structure consisting of a double hollow cylinder, sheath, and a core	No
F1-3	Similar to R pyocins	Non-contractile rod with a uniform width (resemble flexible phage tails)	No
S1-6	DNase, tRNase, Pore-forming, and rRNase activity	A single protein consisting of several functional domains including: receptor-binding domain (I), translocation domain (II), domain with an unknown function (III), and killing domain (IV)	Yes

**Table 2 antibiotics-11-01366-t002:** The mode of action and domains of different S-pyocins.

Pyocin	Mode of Action	Toxin Domains	Immunity Domain
S1	DNase	PF12639 (Colicin-DNase)	PF01320 (Colicin immunity protein)
S2	DNase	PF12639 (Colicin-DNase)	PF01320 (Colicin immunity protein)
S3	DNase	PF06958 (Colicin-DNase)	-
S4	tRNase	PF12106 (Colicin E5 ribonuclease)	PF11480 (Colicin E1 immunity protein)
S5	Pore-forming	PF01024 (Colicin pore-forming domain)	PF03526 (Colicin E1 immunity protein)
S6	rRNase	PF09000 (Cytotoxic)	-
S7	rRNase	PF09000 (Cytotoxic)	-
S8	DNase	PF12639 (DNase/tRNase domain of colicin-like bacteriocin)	PF01320 (Colicin immunity protein)
S9	DNase	PF12639 (DNase/tRNase domain of colicin-like bacteriocin)	PF01320 (Colicin immunity protein)
S10	DNase	PF06958 (S-type Pyocin)	-
S11	tRNase	PF11429 (Colicin D)	PF09204 (Bacterial self-protective colicin-like immunity)
S12	tRNase	PF11429 (Colicin D)	PF09204 (Bacterial self-protective colicin-like immunity)

**Table 3 antibiotics-11-01366-t003:** In vivo models used to assess the effectiveness of different pyocins.

Pyocin	Model	Treatment Route	Protective Effect	Toxicity	References
Crude pyocin (Unknown)	Chick embryos	i.v.	Yes	Yes	Bird et al. [49]
Purified pyocin 78-C2 (strain 78, pyocin type C2)	Murine (LACA strain)	i.v	Yes	No	Merrikin et al. [50]
Crude pyocin	Murine (model not specified)	i.p	Yes	No	Haas et al. [51]
Purified R-type	Murine (CFE mice)	i.p	Yes	No	Rosamu et al. [52]
F-type	Murine (CFE mice)	i.p	Yes	No	Rosamu et al. [52]
S-type	Murine (CFE mice)	i.p	No	NA	Rosamu et al. [52]
Purified R2-pyocin	Murine Peritonitis Model	i.p/i.v	Yes	NA	Scholl et al. [41]
Recombinant pyocin S2	Galleria Mellonella Larvae	N/A	Yes	No	Smith et al. [47]
Recombinant S2 pyocins	Acute *P. aeruginosa* Lung Infection	N/A	Yes	No	Paškevičius et al. [48]

Abbreviations: N/A: not applicable; i.p: intraperitoneal; s.c: subcutaneous; i.v: intravenous; i.n: intranasal.

## Data Availability

Not applicable.

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
