# Peer review of "The In Vivo and In Vitro Assessment of Pyocins in Treating Pseudomonas aeruginosa Infections"

_antibiotics, 2022, doi:10.3390/antibiotics11101366_

Round 1
Reviewer 1 Report
The whole review is well-organized and clear. This reviewer has one major concern regarding the future direction of pyocins in therapeutic and/or clinical applications. What is the bottom line of this review including the authors' standpoint on this topic? Is the title misleading?
Author Response
Reviewer #1
The whole review is well-organized and clear. This reviewer has one major concern regarding the future direction of pyocins in therapeutic and/or clinical applications. What is the bottom line of this review including the authors' standpoint on this topic? Is the title misleading?
We agree with what the reviewer and thank they for the encouraging comments. We changed the title of the manuscript to, “The In-vivo and In-vitro Assessment of Pyocins in Treating P. Aeruginosa Infections”. We did this to emphasize the pre-clinical data of pyocins’ effectiveness against P. aeruginosa infections in this review.
Reviewer 2 Report
Bacteria used several strategies to outcompete other bacteria. One such strategy that bacteria use is bacteriocins, which are antibacterial proteins, secreted by bacteria to kill rival bacteria. In this present review, the authors described the structure, and functions of one such bacteriocins, known as pyocins, found in Pseudomonus aeruginosa. Besides, they also discussed the antibacterial activity of pyocins by referring to several in-vitro and in-vivo studies. Finally, they also argued about the potential use of the pyocins against multidrug-resistant P. aeruginosa. This review is informative and well-written. I have some comments that authors may consider for a better understanding of the review.
1) The authors may consider adding a schematic illustration of the domains found in various pyocins. It will help the reader to understand the difference in the domain architecture present in pyocins groups.
2) I will also suggest to add a diagram explaining the antibacterial activities of various pyocins in bacteria.
Author Response
Reviewer #2
Bacteria used several strategies to outcompete other bacteria. One such strategy that bacteria use is bacteriocins, which are antibacterial proteins, secreted by bacteria to kill rival bacteria. In this present review, the authors described the structure, and functions of one such bacteriocins, known as pyocins, found in Pseudomonus aeruginosa. Besides, they also discussed the antibacterial activity of pyocins by referring to several in-vitro and in-vivo studies. Finally, they also argued about the potential use of the pyocins against multidrug-resistant P. aeruginosa. This review is informative and well-written. I have some comments that authors may consider for a better understanding of the review.
- The authors may consider adding a schematic illustration of the domains found in various pyocins. It will help the reader to understand the difference in the domain architecture present in pyocins groups.
We provided in the revised a manuscript a figure showing the different genes and their corresponding functions for the R2 and F2 loci within the chromosome of the P. aeruginosa PA01. In addition, we have modified table 2 to include the toxicity and immunogenic domains for each of the S-pyocins.
- I will also suggest to add a diagram explaining the antibacterial activities of various pyocins in bacteria.
Table 1 contain information regarding the mode of action of R-, F-, and S-pyocins. In addition, table 2 contains the specific domain in each of the S-pyocins involved in their toxicity. Furthermore, Figure 1 illustrates the genes and the domains of the R2 and F2 pyocins.
Reviewer 3 Report
The present manuscript is a review dedicated to discussing the structure, function and use of piocins in the pathogenesis and treatment of P. aeruginosa infections. This review can be helpful, but there are some concerns.
The most significant problem is the lack of critical discussion and clear commentary on the possible connection of published clinical studies to the subject or, if there is no such connection, highlighting the need for such studies.
Other concerns are regarding formatting, the terms in vivo and in vitro should be in italics.
Author Response
Reviewer #3
The present manuscript is a review dedicated to discussing the structure, function and use of pyocins in the pathogenesis and treatment of P. aeruginosa infections. This review can be helpful, but there are some concerns.The most significant problem is the lack of critical discussion and clear commentary on the possible connection of published clinical studies to the subject or, if there is no such connection, highlighting the need for such studies.
To our knowledge, there are no current clinical studies in which pyocins have systemically examined in humans. We discussed the lack of these studies in the section titled, “Advantage of Pyocins in Treating P. aeuroginsoa Infections”. In the same section, we have highlighted the main advantages of pyocins as potential anti-pseudomonas agents, which includes their differences in the target/mechanism and their receptors. Therefore, utilization of two or more pyocins will significantly reduce the chance of the emergence of anti-pyocin resistant bacteria. This feature illustrates the advantage of pyocin therapy versus bacteriophage therapy.
Other concerns are regarding formatting, the terms in vivo and in vitro should be in italics.
We made the appropriate change to the manuscript.
Round 2
Reviewer 3 Report
The manuscript was revised significantly and may be accepted for publication in present form